# Maternal and Fetal Outcomes of COVID-19 According to the Trimester of Diagnosis: A Cross-Sectional Prospective Study in a Tertiary University Hospital

**DOI:** 10.3390/jcm13175262

**Published:** 2024-09-05

**Authors:** Naser Al-Husban, Rahaf Mohammad Di’bas, Sara Salem Karadsheh, Lara Ali Alananzeh, Iman Aolymat, Asma Kilani, Ala’eddien Obeidat, Alhareth Eid Alhusban, Hedaieh Al-Husban

**Affiliations:** 1Department of Obstetrics and Gynecology, School of Medicine, The University of Jordan, P.O. Box 2194, Amman 11941, Jordan; 2Jordan University Hospital, Amman 11942, Jordan; rahafdibas47@yahoo.com (R.M.D.); karadshehsarah@gmail.com (S.S.K.); ananzeh_l@yahoo.com (L.A.A.); 3Department of Anatomy, Physiology and Biochemistry, Faculty of Medicine, The Hashemite University, P.O. Box 330127, Zarqa 13133, Jordan; imank@hu.edu.jo; 4School of Medicine, The University of Jordan, Amman 11942, Jordan; asma_kilani@yahoo.com (A.K.); alaaobeidat2001@gmail.com (A.O.); alh0211493@ju.edu.jo (A.E.A.); 5Ibn AlHaitham Hospital, Amman 11942, Jordan; husban99@yahoo.com

**Keywords:** COVID-19, fetomaternal, outcome, pregnancy, trimester

## Abstract

**Objectives:** Pregnant women are considered a high-risk group because they may be particularly susceptible to COVID-19. Our study tried to relate fetomaternal outcomes and trimester-specific infection. **Methods:** A prospective study on 224 pregnant women with confirmed antenatal infections at a tertiary hospital. Data from the antenatal clinic records, admission files, labor ward and neonatal notes, lab results, respiratory consultations, and ICU admission were analyzed using Jamovi 2.2.5, with *p* < 0.05 indicating significance. **Results:** A total of 224 patients were included—10, 32, and 182 patients were diagnosed in the first, second, and third trimesters, respectively. Neonatal NICU admissions were significantly higher among those with infections in the third trimester compared to those in the first trimester (*p* = 0.008). Significant differences in Apgar scores at 1 and 5 min emerged between the second and third trimesters (*p* = 0.014 and *p* = 0.037, respectively). However, no significant differences were observed in Apgar scores between the first and second trimesters (*p* = 0.341, *p* = 0.108) or the first and third trimesters (*p* = 0.545, *p* = 0.755). Complications of pregnancy, including maternal mortality and various conditions (respiratory, obstetrical, sepsis, DIC), neonatal outcomes, ICU admission, and cesarean section indications, showed no significant differences related to the trimester of infection (*p*-values: 0.989, 0.892). Study limitations include sample size impacting generalization, higher COVID-19 cases in the third trimester than other trimesters, and potential historical data availability and accuracy issues. **Conclusions:** In the third trimester, COVID-19 caused more neonatal ICU admissions than the first trimester, with lower Apgar scores at 1 and 5 min compared to the second trimester, indicating an increasing susceptibility and vulnerability to COVID-19 infection with an increasing pregnancy age. Other fetal and maternal outcomes showed no significant differences in infection timing.

## 1. Introduction

The coronavirus disease 2019 (COVID-19) is caused by a single-stranded RNA-enveloped virus, which causes a wide range of symptoms with an incubation period varying from 2 to 12 days.

It began in Wuhan, China, in December 2019, and since then, the number of confirmed cases and associated mortality and morbidity have increased rapidly. By 12 March 2020, the World Health Organization (WHO) classified the outbreak as a pandemic [1]. This pandemic negatively affected many people, including pregnant women and their fetuses, increasing the maternal death rates from 5.17 per 100,000 pregnancies in the year prior to the pandemic to 8.69 fatalities per 100,000 during the pandemic [2]. There were also nationwide lockdowns, disruption of healthcare services, and fear of attending healthcare facilities that affected the well-being of pregnant people and their babies, thus increasing maternal mental health issues and rates of stillbirth and preterm birth, especially in low- and middle-income countries [3,4].

### 1.1. Physiological and Immunological Vulnerabilities of Pregnant Women to COVID-19

Pregnant women are considered a high-risk group because they may be particularly susceptible to COVID-19 due to the physiologic changes of pregnancy involving the cardiorespiratory system in which the enlarged uterus raises abdominal pressure and lifts the diaphragm. Also, hormonal changes increase the subcostal angle and outward expansion of the rib cage while decreasing chest wall compliance, leading to “physiologic dyspnea” in many pregnant women and decreasing functional residual capacity, resulting in compensatory respiratory alkalemia. This change can shift the oxygen dissociation curve to the right, which is beneficial in transferring oxygen to the fetus but may hinder respiratory function in pregnant women with pulmonary complications resulting in altered response to COVID-19 infection in pregnancy, with the third trimester being the riskiest because the heavily pregnant uterus compresses the lungs more than in early pregnancy [2,5].

Another system is the immune system; pregnancy is an immunologically vulnerable period because of the dilutional anemia that happens in 38% of pregnant women, leading to reduced blood oxygen content, prioritizing vital organs like the heart and brain over immune organs, potentially compromising immune function. Also, hormonal changes, particularly human chronic gonadotropin and progesterone, modulate immune responses and fetal tolerance. These factors result in placing pregnant women at risk for more severe illness and complications from infections [5].

### 1.2. Impact of COVID-19-Induced Cytokine Storm on Fetomaternal Outcomes

It has been reported that COVID-19, especially in severe cases, can lead to a cytokine storm. Additionally, pregnant women in their first and third trimesters of pregnancy are in a pro-inflammatory state due to the complex interaction between the immune system and sex hormones. Therefore, the cytokine storm induced by SARS-CoV-2 in these periods may exacerbate the severity of the inflammatory state, potentially resulting in adverse obstetric outcomes [5].

Even without a placental infection, raised inflammatory cytokines levels were seen in the neonates’ cord blood. It is still not obvious if these cytokines were fetal or maternal in origin. Immune cells in cord blood were shown in research to exhibit higher cytokine production in SARS-CoV-2-infected pregnancies [6]. 

### 1.3. COVID-19 Impact on the Placenta 

Placental infection by COVID-19 appears to be uncommon. Even without placental infection, CoVID-2-related inflammation and coagulation still occur and manifest most commonly as thrombosis and fibrin deposition in the intervillous spaces [6]. COVID-19 increases the risk of thromboembolic complications in the general population, and while pregnancy is, by itself, a hypercoagulable state with increased thrombin production and an increase in intravascular inflammation, COVID-19 increases the risk of preeclampsia, preterm birth, and other adverse pregnancy outcomes as an additive or synergistic risk factor for thrombosis during pregnancy, leading to uteroplacental malperfusion. To avoid these events, the RCOG recommended that all pregnant women admitted with confirmed COVID-19 should receive prophylactic low-molecular-weight heparin (LMWH) [7,8].

### 1.4. The Increased Risk of Complications Associated with COVID-19 during Pregnancy

Taking all of this into consideration, it is understandable why pregnant individuals were found to be at a heightened risk of more severe symptoms than people who were not pregnant. One study showed that the risk of adult respiratory distress syndrome and acute renal failure was between 1.66 and 2.22 times higher in mothers with COVID-19 infections during pregnancy compared to those without [9], and also were more likely to be admitted to an intensive care unit (4%), require invasive ventilation [10], need a C-section (45.4 versus 32.4 percent), deliver preterm (26.9 versus 14.1 percent), die around the time of birth, and experience postpartum hemorrhage [11]. Other recent studies concluded that severe COVID-19 is strongly associated with preeclampsia, gestational diabetes, preterm birth, and low birth weight [12].

### 1.5. Vertical Transmission and Breastfeeding Considerations in COVID-19

Fortunately, the rate of vertical transmission of the virus to the fetus transplacentally reported in women with COVID-19 proved to be a minority, and most of the cases of neonatal SARS-CoV-2 are due to postnatal infection [13,14]. This could be because the virus is known to have a very low level of viremia, and the level of virus receptor expression in the placenta that facilitates viral entry is very low [12]. 

In addition, breastmilk transmission was unlikely [3]. In contrast, when breastfeeding, secretory IgA antibodies and other immunologically active substances in breast milk are passed to the suckling infant, supporting their immature adaptive immune system. Therefore, breastfeeding by infected mothers was encouraged [12]. 

### 1.6. COVID-19 Vaccination in Pregnancy: Safety, Efficacy, and Fetal Outcomes

In pregnancy, COVID-19 vaccine adverse events have not been shown to be different from those of the general population. In addition, no adverse outcomes were reported for obstetric, fetal, or neonatal patients [12]. On the contrary, they found that severe complications, such as critical care admission, stillbirths, and early neonatal deaths, were more common in those who were unvaccinated compared with those who were vaccinated [15] and that there is 89% vaccine effectiveness against COVID-19 hospitalizations when given during pregnancy [2]. As for the effect of vaccination on fetal outcomes, it was found that the risk of infant hospitalization for COVID-19 up to 4–6 months of age could be reduced by 30–70% by vaccinating pregnant women [12].

## 2. Methods and Population

### 2.1. Study Population

Pregnant women who have had a confirmed COVID-19 infection by nasopharyngeal swab(s) at a University Hospital in the period February 2020–August 2022. These women were infected in different trimesters of pregnancy.

### 2.2. Follow-up 

They were followed from the time of infection until 6 weeks postpartum. They were assessed regarding hospitalization due to the infection and/or its complications, sepsis, death, respiratory complications, disseminated intravascular coagulation (DIC), ICU admission, antenatal clinic visits, miscarriages, preterm deliveries, development of pre-eclampsia, induction of labor, mode of delivery, indications for cesarean sections (CS), complications of delivery, intrauterine growth restriction (IUGR), stillbirths, neonatal intensive care unit (NICU) admission and its outcome, and breastfeeding practices in these women.

### 2.3. Data Collection and Sources

The obstetric outcomes were obtained from the antenatal clinic and admission notes, labor ward data, postnatal clinic notes, or admissions. The respiratory variables were obtained from respiratory consultations and interventions, anesthesia notes and the intensive care unit admissions notes. The fetal variables were obtained from the nursery and neonatal intensive care unit notes and neonatologists’ notes.

### 2.4. Inclusion and Exclusion Criteria

The inclusion criteria included all those who had a confirmed infection and completed data.

The exclusion criteria included women with multiple pregnancies, those with suspected but not confirmed infections, those with chronic underlying medical disorders, and those with missing data.

### 2.5. Ethical Considerations and Registration

Ethics approval for this study was obtained from the institutional review board (IRB) at Jordan University Hospital (Decision number 307/2020, dated 29 December 2020). The study involved a prospective review of obstetric, medical, and respiratory clinical data and notes collected over a specified period. As the data did not include any personal, individual, or identifying information, written informed consent was not required. Throughout the study, all information was anonymized, and strict measures were implemented to ensure participant confidentiality. No identifying data will be shared.

The study was registered in clinicaltrials.gov with a ClinicalTrials.gov Identifier: NCT04869202.

This study has been reported in line with the STROCSS criteria [16].

### 2.6. Statistical Analysis

Data were entered into and analyzed using Jamovi 2.2.5. *p* ≤ 0.05 was considered significant. Data was assessed for normality using Kolmogorov–Smirnov test.

Descriptive analysis was applied to calculate the frequencies and percentages for categorical variables and the means (standard deviations) and medians (interquartile ranges) for continuous variables according to the normality of distribution.

The Chi-squared (2) test was used to test for association between categorical variables.

The Mann–Whitney test was used to find if there are significant differences in gestational age at delivery, birth weight, and Apgar scores between first and second, first and third, and second and third trimesters. 

The Kruskal–Wallis test was used to examine the differences in Apgar scores and birth weight and gestational age (GA) at delivery in relation to the trimester in which the patient became infected.

### 2.7. Definitions

Preeclampsia is characterized by new-onset hypertension after 20 weeks of gestation, defined by a systolic blood pressure of 140 mm Hg or more or diastolic blood pressure of 90 mm Hg or more on two occasions at least 4 h apart or more severe values (diastolic ≥ 110 mm Hg or systolic ≥ 160 mm Hg) [15].

Fetal death, according to the United States Center for Health Statistics, is the delivery of a fetus showing no signs of life, regardless of pregnancy duration. Stillbirth is defined as fetal death at or beyond 20 weeks gestation or with a birth weight of 350 g or more, aiming to standardize definitions [17].

Early pregnancy loss is now described using terms like “miscarriage” or “intrauterine pregnancy loss” for pregnancies lost before 20 weeks gestation, reflecting a shift toward clearer and less stigmatized terminology in medical practice [18,19].

The aim of this study is to assess the fetomaternal outcomes of pregnancies complicated by COVID-19 infection according to trimester at diagnosis in a tertiary university hospital. 

## 3. Results

### 3.1. The Study Population Demographics

The study’s significant contribution lies in its utilization of a distinct dataset sourced specifically from Jordan. This unique dataset offers valuable insights that are contextually relevant to Jordan, thereby enhancing the significance and applicability of the study findings within this regional setting.

Among the participants, a total of 224 patients who had confirmed COVID-19 during pregnancy were included. Most patients (182) were diagnosed in the third trimester.

A detailed description of the study population is shown in Table 1 and Table 2.

### 3.2. Complications of Pregnancy 

There was no significant difference in complications of pregnancy, including maternal mortality, respiratory conditions (pneumonia, hypoxia, respiratory failure, pulmonary hypertension), obstetrical conditions (severe preeclampsia, oligohydramnios, gestational hypertension, uncontrolled gestational diabetes, and placenta accreta), sepsis, DIC and ICU admission in relation to the trimester in which the patient became infected (*p* = 0.989) (Table 2). Perhaps the overall fetomaternal complications in our study were low because we excluded patients with comorbidities such as hypertension, diabetes mellitus, lupus, nephropathy, antiphospholipid antibody syndrome, and others.

### 3.3. Indications for CS

There was no significant difference in indications for CS, including elective, fetal distress, IUGR, maternal respiratory conditions, and oligohydramnios, in relation to the trimester in which the patient became infected (*p* = 0.892) (Table 2). A total of 86% of CS was done under spinal anesthesia.

### 3.4. Neonatal Outcome

There was a significant difference in the neonatal admission to the NICU (*p* = 0.008) in relation to the gestational period at diagnosis. The indications for these admissions were divided into COVID-related and non-COVID-related categories.

### 3.5. Apgar Score

There were significant differences in Apgar scores at 1 and 5 min after birth in relation to the trimester at which the patient contracted COVID-19 infection, with *p*-values of 0.017 and 0.037, respectively.

The percentages of infants with low Apgar scores (≤5) at 1 and 5 min varied by trimester: In the first trimester, 10% (1/10) had low Apgar scores at 1 min, and 0% (0/10) at 5 min. In the second trimester, 12.5% (4/32) had low Apgar scores at 1 min and 6.3% (2/32) at 5 min. In the third trimester, 2.2% (4/182) had low Apgar scores at 1 min and 0.5% (1/182) at 5 min.

There was a significant difference in Apgar 1 and 5 between the second and third trimesters (*p* = 0.014 and *p* = 0.037, respectively). However, there were no significant differences in Apgar 1 and 5 between the first and second trimesters (*p* = 0.509, *p* = 0.321) or the first and third trimesters (*p* = 0.480, *p* = 0.683) (Table 3).

### 3.6. GA at Delivery and Birth Weight

There were no significant differences in GA at delivery or birth weight in relation to the trimester in which the patient became infected with COVID-19; *p*-values are 0.829 and 0.304, respectively. The Mann–Whitney test revealed no significant differences in birth weight and GA at delivery between the first, second, and third trimesters, as shown in (Table 3).

## 4. Discussion

Many data and recommendations have changed regarding guidelines, virus profile and administration of patients and pregnant women with COVID-19. All patients in our study received one or another form of treatment ranging from nothing to decongestants and antipyretics to the extreme situation of antiviral, steroids and ventilatory support.

As there were very little data considering the effects of COVID-19 infection in relation to the trimester of infection and a robust collection of maternal data by trimester of exposure, including the periconception period, was required to determine these effects, we conducted such a study.

### 4.1. Complications of Pregnancy and Maternal Outcomes

Infection during the third trimester poses a higher risk for ICU admission, critical illness, mechanical ventilation, and death [20,21]. A recent study observed higher maternal and fetal complications in women diagnosed in the third trimester with higher maternal mortality rates and a greater frequency of pregnancy-induced hypertension compared to other trimesters [22]. On the other hand, a study of 114 infected women discovered that higher GA was protective against severe illness [23]. Our findings align with this study, revealing that advanced gestational infection does not carry more adverse maternal outcomes than early infection. Hypertension amplifies the risk of death in the third trimester [24]. However, we excluded mothers with co-existing medical disorders such as hypertension from our study to ensure consistency in the study population and rule out contributing factors to adverse fetomaternal outcomes.

A study showed that women with active infections by the time of delivery had a significantly higher rate of hospital admission for COVID-19 and ICU admission than the recovered group [25]. We did not specifically study those cases with an active, symptomatic intrapartum infection.

UK data showed that early pregnant women with COVID-19 represented a minority of hospitalized pregnant women [21]. Maternal outcomes were favorable when infection occurred in the first trimester, and women who experienced spontaneous abortion in the first trimester had similar antibody prevalence to those with ongoing pregnancies [26]. The minimal respiratory changes during this phase may explain this. However, first-trimester infection could still lead to complications due to SARS-CoV-2 invasion of the placental villi in subsequent weeks of gestation [27]. A study found that there was a higher proportion of placentas from patients with COVID-19 in earlier gestations with evidence of placental infection-associated features [6].

It was found in small case reports of COVID-19-infected pregnant women that no miscarriages occurred in the first trimester of pregnancy and, in fact, they all were second or third-trimester miscarriages [28]. Meanwhile, in asymptomatic pregnant women, the COVID-19 environment did not appear to influence rates of first-trimester miscarriage [29]. On the other hand, research suggested that miscarriage was more commonly seen in patients who became ill in the first trimester compared to the second trimester [20].

One study revealed a significant association between miscarriage and trimester at diagnosis of infection but no significant association between stillbirth, pre-eclampsia, or trimester at diagnosis [30]. Our study had the same result concerning stillbirth and pre-eclampsia: we found no significant association between complications of pregnancy that included maternal mortality, obstetrical conditions involving pre-eclampsia, ICU admission, intrauterine fetal demise, and trimester at diagnosis (*p* = 0.989) (Table 2).

### 4.2. GA at Delivery and NICU Admission

Fetal loss during the first 24 weeks of gestation was due to several systematic inflammatory events, including the placenta with subsequent premature labor, rupture of membranes, or placental insufficiency [31,32].

Among infants born to mothers who tested positive 0–14 days before delivery, more were admitted to the NICU than those born to mothers who tested positive more than 14 days before delivery and were also born earlier [33]. We did not examine the impact of infection timing in the third trimester on fetomaternal outcomes. Prematurity may be due to maternal complications, intrauterine infection, the perceived need for early antiviral treatment in the disease course [34], or it may be iatrogenic [35].

Our study found no significant association between GA at diagnosis and GA at delivery, as well as NICU admission, which is consistent with previous research [30]. However, we did find a significant association between the trimester at diagnosis and NICU admission (*p* = 0.008), which is an interesting finding. That could be explained by the fact that most of the neonatal complications in COVID-19-positive mothers were due to prematurity and not COVID-19 infection. A multicenter study showed that if mothers were admitted for COVID-19 infection, their newborns had a higher risk of premature delivery [36]. Women with COVID-19 infection in their third trimester had the highest frequencies of preterm labor and are at higher risk of adverse outcomes [20,22], which increases the likelihood of NICU admission. All these facts together could explain the results of our study, which found an association between NICU admission and trimester at diagnosis.

There was a lack of information on the vertical transmission potential, resulting in understandable anxiety among women and obstetricians, which in turn was reflected in the high rates of preterm birth [21], which was similarly explained in our study. The evidence inferred that vertical transmission of SARS-CoV-2 was rare [37]. In our study, the number of neonates born to COVID-19-infected mothers was zero, and similarly, no cases of vertical or horizontal transmission were found in another study [36]. The chance that congenital anomalies were caused by COVID-19 was low [37].

A decline in the number of NICU admissions and neonatal resuscitations during the pandemic was seen, which might have been explained by travel bans and strict measures of infection prevention and control [38]. In our study, the indications for admission were 41.4% COVID-related and 58.6% non-COVID-related.

### 4.3. Apgar Score

Our study found that neonates whose mothers were diagnosed with infections in the second trimester had significantly different Apgar scores at 1 and 5 min compared to those diagnosed in the third trimester. However, there were no significant differences in Apgar scores between the first and second or first and third trimesters. In a recent study, it was observed that women with COVID-19 infection in the third trimester had the lowest Apgar scores and highest in patients with COVID-19 infection in the first trimester [22]. We could not draw clear conclusions about the risk of low Apgar scores due to insufficient evidence. It is possible that pregnant women experience a transition to a pro-inflammatory state in the third trimester and an anti-inflammatory state in the second trimester. Given the association of COVID-19 with cytokine storms, pregnant women infected during the third trimester might exhibit reduced Apgar scores due to the heightened inflammatory response, potentially impacting fetal health [5]. Contrary to expectations, there were no significant differences in Apgar scores between the first and third trimesters, possibly due to the small number of pregnant women diagnosed in the first trimester (*n* = 10) compared to the third trimester (*n* = 182). Larger studies are needed to establish the true association.

### 4.4. Birth Weight

The pathological placental features, including multiple villous infarcts, could result in subsequent miscarriage or IUGR [32]. Placental hypoperfusion was associated with occlusive fibrin deposition and non-occlusive thrombi [21,30]. The small for gestational age (SGA) risk was found to be comparable across all trimesters [39]. Another study revealed that there was no significant difference in birthweight, birthweight percentile, or rates of SGA infants (5.95% in the active-infection group, which was low when compared to 8.75% in the recovered group) [25]. Similarly, there was no significant association between IUGR and trimester at diagnosis [30]. These findings were consistent with our study, as we did not find a significant association between birth weight and trimester at diagnosis (*p* = 0.304).

### 4.5. Mode of Delivery 

During the initial stages of the pandemic in China, 90% of infected women were delivered by CS [36], which might indicate that obstetricians lack enough knowledge about the effects of COVID-19 on pregnancy or that cesarean was necessary because of worsening maternal symptoms [35]. Those who required hospitalization had a high cesarean section rate. Following current obstetric guidelines over time, indications for CS were gradually reduced to scenarios with maternal or fetal compromise [36]. The rate of CS in our study was 50%, with elective indications being the highest and oligohydramnios being the least. Similar to our findings, there was no significant association between the mode of delivery and the trimester of infection diagnosis [25].

### 4.6. Breastfeeding 

WHO and CDC recommended the wearing of a facemask and continued breastfeeding by infected mothers. While one study reported finding the virus in breastmilk one week after delivery, two small studies found no molecular evidence of the virus in breastmilk [26].

Providing support to breastfeeding mothers was very important, as this infection was a major cause of anxiety [35]. Fortunately, the rate of mothers breastfeeding in our study was 78.1%. This might be partly due to the strong societal and cultural ties and partly to the early widespread vaccination, which caused relatively mild infection in most cases and decreased anxiety in pregnant women.

### 4.7. Limitations

There are some potential limitations to our study. It is a single-centered study. The sample size of the study may be limited, potentially reducing the generalization of the findings to a broader population. The number of pregnant women who became infected with COVID-19 in the third trimester is much higher than in the second and first trimesters. Lastly, there may be limitations in terms of the availability and accuracy of historical data. This could introduce errors in the documentation of events and outcomes.

It is important to differentiate this kind of infection from other infections like ZIKA VIRUS, considering the relatively recent discovery of this congenital infectious syndrome. Further studies and updated long-term follow-up are needed [40]. In Jordan, termination of pregnancy is totally prohibited and illegal unless the pregnancy is threatening the mother’s life. In our hospital, there were no pregnancy terminations due to COVID infection, and there was no evidence of vertical transmission [41]. In our hospital, we did not study the different variants of the virus. The delta variant was found to be associated with more adverse fetomaternal outcomes than other variants [42]. Around the same period of time of our study, it was found that there were moderate scores in COVID-19 vaccine hesitancy among Jordanian pregnant women despite the current international recommendations for its safety for women and their fetuses or neonates [43]. Moreover, it was found that about a third of pregnant women were still hesitant about the vaccine, probably because of the conflicting information [44].

## 5. Conclusions 

In the third trimester, COVID-19 caused more neonatal ICU admissions than in the first trimester, with lower Apgar scores at 1 and 5 min compared to the second trimester, indicating an increasing susceptibility and vulnerability to COVID-19 infection with an increasing pregnancy age. Other fetal and maternal outcomes showed no significant differences in infection timing.

## Figures and Tables

**Table 1 jcm-13-05262-t001:** An overview of the means (standard deviations) and medians (interquartile ranges) according to the normality of distribution and probability values for several variables in relation to trimester at the time of diagnosis.

	First Trimester	Second Trimester	Third Trimester	* p *-Value
Age	32.4 (SD 5.19)	32.1 (SD 5.14)	31 (IQR 27.75–35)	0.51
GA at delivery	38.1 (SD 1.10)	38 (IQR 36.25–40)	38 (IQR 36–39)	0.829
Birth weight	3.11 (SD 0.48)	2.85 (IQR 2.56–3.1)	3 (IQR 2.7–3.26)	0.304
Apgar (min 1)	8 (IQR 8–8)	8 (IQR 8–8)	8 (IQR 8–8)	0.017
Apgar (min 5)	9 (IQR 9–9)	9 (IQR 9–9)	9 (IQR 9–9)	0.037

*p*-value; probability value.

**Table 2 jcm-13-05262-t002:** An overview of the frequencies, percentages, and probability values for several variables in relation to the gestational period at diagnosis.

	* N *	%	First	Second	Third	*p*-Value
			(*N =* 10)	(*N =* 32)	(*N =* 182)	
Parity	224					
Nulliparity	1	0.4%	0/10	0/32	1/182	
Multiparity	152	67.9%	5/10	16/32	131/182	
Grand multiparity	71	31.7%	5/10	16/32	50/182	
Complications	39					* p = * 0.989
Death	2	5.1%	0/1	0/7	2/31	* p = * 0.762
Sepsis	2	5.1%	0/1	0/7	2/31	* p = * 0.762
Respiratory	7	17.9%	0/1	0/7	7/31	* p = * 0.333
Obstetric	16	41%	0/1	3/7	13/31	* p = * 0.699
Neonatal	18	46.2%	1/1	5/7	12/31	* p = * 0.161
DIC	3	7.7%	0/1	0/7	3/31	* p = * 0.657
ICU	5	12.8%	0/1	0/7	5/31	* p = * 0.477
NICU admission	224					* p = * 0.008
Dead	4	1.8%	0/10	3/32	1/182	
Admitted to the NICU	58	25.9%	1/10	7/32	50/182	
Not admitted to the NICU	162	72.3%	9/10	22/32	131/182	
Indications of NICU admission	58					* p = * 0.387
Non-COVID-related	34	58.6%	0/1	5/7	29/50	
COVID-related	24	41.4%	1/1	2/7	21/50	
Mode of delivery	224					* p = * 0.391
Normal vaginal delivery	112	50%	6/10	19/32	87/182	
Cesarean section	112	50%	4/10	13/32	95/182	
Indications of Caesarean section	112					* p = * 0.892
IUGR	5	4.5%	0/4	0/13	5/95	* p = * 0.626
Oligohydramnios	1	0.9%	0/4	0/13	1/95	* p = * 0.914
Maternal respiratory condition	3	2.7%	0/4	0/13	3/95	* p = * 0.759
Fetal distress	14	12.5%	¼	3/13	10/95	* p = * 0.326
Elective	89	79.5%	¾	10/13	76/95	* p = * 0.943
Breastfeeding	224					* p = * 0.604
Yes	175	78.1%	9/10	24/32	142/182	
No	49	21.9%	1/10	8/32	40/182	

*N* is the number of non-missing values. *p*-value—probability value.

**Table 3 jcm-13-05262-t003:** The probability values of comparing the trimesters in which the patient contracted COVID.

*p*-Value					
Trimester at Time of Diagnosis		Birth Weight	Gestational Age at Delivery	Apgar (1 min)	Apgar (5 min)
First	Second	0.183	0.695	0.509	0.321
First	Third	0.276	0.539	0.48	0.683
Second	Third	0.299	0.877	0.014	0.037

*p*-value—probability value.

## Data Availability

All data are available from the corresponding author on reasonable request.

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
