# Peer review of "Maternal and Fetal Outcomes of COVID-19 According to the Trimester of Diagnosis: A Cross-Sectional Prospective Study in a Tertiary University Hospital"

_jcm, 2024, doi:10.3390/jcm13175262_

Round 1

Reviewer 1 Report

Comments and Suggestions for Authors

The Authors analyze maternal and fetal impact of COVID-19 infection in a prospective study conducted in a tertiary university hospital. They declare, that the presented data is specific to Jordan, offering a valuable contextual insight.

An increased maternal morbidity of COVID-19 was demonstrated with progressing pregnancy age. The neonates after infection in the third trimester had lower Apgar scores and required more intensive care than neonates from pregnancies infected in an earlier age.

This study shows, that COVID-19 infecion in advanced pregnancy is bound to more complications than COVID-19 infection in early pregnancy indicating an increasing susceptibility and vulnerability to COVID-19 infection with an increasing pregnancy age.

This paper requires extensive revision, with assistance of a medical English speaker. A reference to the Table 1 should be removed from the abstract. A Paragraph 3.2 is misleading suggesting no impact of COVID-19 infection time point (1st versus 3rd trimster) on fetal outcome, whereas there were significantly lower Apgar scores and more NICU admissions observed in babies born after late pregnancy infection. This information should be clearly stated. The references should be quoted in order of appaerance in the text.

This study adds a limited new data to our knowledge on the the COVID-19 infection in pregnancy.

Comments on the Quality of English Language

The English language needs improvements with assistance of a medical English speaker.

Author Response

I would take this opportunity to thank you for the valuable comments, suggestions and recommendations.

They really significantly improved the overall quality of the manuscript.

All amendments and revision points are shown in track changes.

Reviewer 1 comments;

Comment; The Authors analyze maternal and fetal impact of COVID-19 infection in a prospective study conducted in a tertiary university hospital. They declare, that the presented data is specific to Jordan, offering a valuable contextual insight.

An increased maternal morbidity of COVID-19 was demonstrated with progressing pregnancy age. The neonates after infection in the third trimester had lower Apgar scores and required more intensive care than neonates from pregnancies infected in an earlier age.

This study shows, that COVID-19 infection in advanced pregnancy is bound to more complications than COVID-19 infection in early pregnancy indicating an increasing susceptibility and vulnerability to COVID-19 infection with an increasing pregnancy age.

Response; Many thanks. In fact, based on your statements, we revised our conclusion and quoted ’indicating an increasing susceptibility and vulnerability to COVID-19 infection with an increasing pregnancy age’.

Comment; This paper requires extensive revision, with assistance of a medical English speaker.

Response; Thank you very much. The manuscript was revised by Cambridge-graduate English Speaker.

Comment; A reference to the Table 1 should be removed from the abstract.

Response; It was removed as advised.

Comment; A Paragraph 3.2 is misleading suggesting no impact of COVID-19 infection time point (1st versus 3rd trimester) on fetal outcome, whereas there were significantly lower Apgar scores and more NICU admissions observed in babies born after late pregnancy infection. This information should be clearly stated.

Response; Many thanks for the important note. An amendment was made accordingly.

Comment; The references should be quoted in order of appearance in the text.

Response; we completely agree with this recommendation. However, there are few instances where we could not rearrange them due to the flow of the essential information within the manuscript and the appearance of the cited data in the references (some are different data from the same reference). Anyhow, most of the references are numbered according to their appearance within the manuscript.

Comment; This study adds a limited new data to our knowledge on the the COVID-19 infection in pregnancy.

Response; That is right due to the large data published around COVID-19 infection. However, data relating fetomaternal outcomes to the time of infection in pregnancy in our region generally and in Jordan particularly are not enormous.

Comments on the Quality of English Language; the English language needs improvements with assistance of a medical English speaker.

Response; Thank you very much. The manuscript was revised by Cambridge-graduate English Speaker.

Reviewer 2 Report

Comments and Suggestions for Authors

The most significant problem with any article about COVID is the novelty of the study. I just queried PUBMED and got 430,332 hits when I put in the search term "COVID". Another significant problem is that if you are looking at maternal and fetal outcomes is that in this study you are really only looking at an assessment of the third trimester with 10,32,182 patients in the first second and third trimesters of pregnancy. You can take a very limited look at maternal outcomes with the small numbers in the first and second trimesters of pregnancy, but neonatal outcomes are extremely limited and are overwhelmingly complicated by prematurity in the second trimester and can not be evaluated in the first trimester except to be classified as a fetal loss or not. . 

Introduction: Not sure how helpful sections 1.1 to 1.7 in the introduction are for this paper, and section 1.7 are definitions and if used should probably be in the methods section.  In your results section you use Apgar scores, would use blood gases if they are available, If not average and IQR are also not very helpful. I minute Apgar score informs about the condition of the neonate at birth but 5 minute score is most meaningful and what information is needed is how many of those scores at 5 minutes are 5 or less. 

Comments on the Quality of English Language

English language is acceptable

Author Response

I would take this opportunity to thank you for the valuable comments, suggestions and recommendations.

They really significantly improved the overall quality of the manuscript.

All amendments and revision points are shown in track changes.

Reviewer 2 comments;

Comments and Suggestions for Authors;

The most significant problem with any article about COVID is the novelty of the study. I just queried PUBMED and got 430,332 hits when I put in the search term "COVID". Another significant problem is that if you are looking at maternal and fetal outcomes is that in this study you are really only looking at an assessment of the third trimester with 10,32,182 patients in the first second and third trimesters of pregnancy. You can take a very limited look at maternal outcomes with the small numbers in the first and second trimesters of pregnancy, but neonatal outcomes are extremely limited and are overwhelmingly complicated by prematurity in the second trimester and can not be evaluated in the first trimester except to be classified as a fetal loss or not.

 Response; Unfortunately, that is true. It is due to the fact that there are some potential limitations to our study. It is a single-centered study. The sample size of the study may be limited, potentially reducing the generalization of the findings to a broader population. The number of pregnant women who got their covid-19 in third trimester is much higher than second and first trimesters. These are mentioned in the limitations of the study section.

Comment; Introduction: Not sure how helpful sections 1.1 to 1.7 in the introduction are for this paper

Response; We really want to appreciate this comment.These are reviews of the COVID-19 infection nature with an emphasis and focus on the specifications in pregnancy physiology, its effect on mother, fetus and placenta. We also mention a brief about vaccination in pregnancy. We appreciate that in case you strongly want them to be deleted, we are going to be happy with that decision.

Comment; section 1.7 are definitions and if used should probably be in the methods section. 

Response; Many thanks for that. We have moved definitions to the methods section as advised and given number 2.7.

Comment; In your results section you use Apgar scores, would use blood gases if they are available, If not average and IQR are also not very helpful. I minute Apgar score informs about the condition of the neonate at birth but 5 minute score is most meaningful and what information is needed is how many of those scores at 5 minutes are 5 or less. 

Response; unfortunately, blood gases are not available. All cases have Apgar scores at 1 and 5 minutes.

Comments on the Quality of English Language; English language is acceptable

Response; Thank you very much.

Round 2

Reviewer 1 Report

Comments and Suggestions for Authors

The Authors have improved this manuscript as indicated. I have no other comments.

Comments on the Quality of English Language

They elaborated the references, as well as a stream of the text.

Author Response

Comments and Suggestions for Authors

The Authors have improved this manuscript as indicated. I have no other comments.

Comments on the Quality of English Language

They elaborated the references, as well as a stream of the text.

Response; many thanks. We greatly appreciate your time and efforts in improving the quality of our manuscript

Reviewer 2 Report

Comments and Suggestions for Authors

Unfortunately, the small sample size for pregnancy outcomes and the lack of novelty still makes this study sub optimal for publication. There are now > 438,000 results if you search Covid 19 in PUBMED

Comments on the Quality of English Language

Overall English language is acceptable

Author Response

Comments and Suggestions for Authors

Unfortunately, the small sample size for pregnancy outcomes and the lack of novelty still makes this study sub optimal for publication. There are now > 438,000 results if you search Covid 19 in PUBMED

Response; thank you very much for raising this issue. We mentioned that in the manuscript and indicated these as limitations of our study. 

We have mentioned'

There are some potential limitations to our study. It is a single-centered study. The sample size of the study may be limited, potentially reducing the generalization of the findings to a broader population. The number of pregnant women who got their covid-19 in third trimester is much higher than second and first trimesters. Lastly, there may be limitations in terms of the availability and accuracy of historical data. This could introduce errors in the documentation of events and outcomes'.

Comments on the Quality of English Language  

Overall English language is acceptable

Response; thank you very much. Your time and effort improving our manuscript overall quality are greatly appreciated.